

# Self voting classification model for online meeting app review sentiment analysis and topic modeling

Naila Aslam[1], Kewen Xia[1], Furqan Rustam[2], Ernesto Lee[3] and Imran Ashraf[4]

[1] School of Electronics and Information Engineering, Hebei University of Technology, Tianjin, China
[2] School of Computer Science University College Dublin, Dublin, Ireland
[3] College of Engineering and Technology Miami Dade College, Miami, FL, USA
[4] Information and Communication Engineering, Yeungnam University, Gyeongsan-si, Republic of Korea

Corresponding authors
Kewen Xia, kwxia@hebut.edu.cn
Imran Ashraf, imranashraf@ynu.ac.kr

## ABSTRACT

Online meeting applications (apps) have emerged as a potential solution for conferencing, education and meetings, etc. during the COVID-19 outbreak and are used by private companies and governments alike. A large number of such apps compete with each other by providing a different set of functions towards users' satisfaction. These apps take users' feedback in the form of opinions and reviews which are later used to improve the quality of services. Sentiment analysis serves as the key function to obtain and analyze users' sentiments from the posted feedback indicating the importance of efficient and accurate sentiment analysis. This study proposes the novel idea of self voting classification (SVC) where multiple variants of the same model are trained using different feature extraction approaches and the final prediction is based on the ensemble of these variants. For experiments, the data collected from the Google Play store for online meeting apps were used. Primarily, the focus of this study is to use a support vector machine (SVM) with the proposed SVC approach using both soft voting (SV) and hard voting (HV) criteria, however, decision tree, logistic regression, and k nearest neighbor have also been investigated for performance appraisal. Three variants of models are trained on a bag of words, term frequency-inverse document frequency, and hashing features to make the ensemble. Experimental results indicate that the proposed SVC approach can elevate the performance of traditional machine learning models substantially. The SVM obtains 1.00 and 0.98 accuracy scores, using HV and SV criteria, respectively when used with the proposed SVC approach. Topic-wise sentiment analysis using the latent Dirichlet allocation technique is performed as well for topic modeling.

## INTRODUCTION

Online meeting applications (apps) have emerged as a potential solution for meetings, online education, discussion forums, *etc.*, during the COVID-19 pandemic. Many companies and governments alike initiated the concept of working from home. Similarly,

educational institutes started remote classes online, business meetings were organized virtually and this has become possible using online meeting apps such as Google Meet, Zoom, Microsoft team viewer, *etc.*, Reports show that 75% of employees depend on online video conference technology amid the COVID-19 pandemic (*Spotme, 2021*). Similarly, 30% travel expenses have been dropped down and 11,000 US dollars (USD) have been saved by companies per employee using these online video conference plate forms (*Spotme, 2021*). Online meeting apps have been presented both for computers and mobile devices, the major part of which constitutes smartphones. A large number of online meeting apps are available on the Google Play store and new apps are begin contrived and developed by different companies. The rise in the development of meeting apps is attributed to significant growth of 8.1% in 2020 amid the traveling and office working constraints during the COVID-19 outbreak (*Fortune Business Insights, 2022*). This growth is expected to reach a total of 12.99 billion USD by 2028 which is currently 6.28 billion USD (*Fortune Business Insights, 2022*).

Available online meeting apps provide a rich variety of functions to facilitate online meetings, however, such apps are not without their demerits which often come from the bugs in the app programming. Similarly, the level of satisfaction for one app varies from the other regarding user-friendliness, functions, cost, *etc.*, User gives reviews about apps' features and discusses the issues they face while using such apps. Such reviews/opinions contain the sentiments of users and are helpful to point out the limitations and suggest additional features to increase the level of quality and user satisfaction. However, finding and prioritizing such views require a systematic analysis of the app's reviews using a suitable approach.

This study presents a systematic approach to perform sentiment analysis and topic modeling of online meeting app reviews to find people's opinions regarding the use of such apps. For this purpose, a supervised machine learning framework is utilized and the following contributions are made

- The study performs sentiment analysis of tweets for online meeting apps using a novel self-voting ensemble model. The self-voting model combines three variants of the same model, however, the features fed to each model are different. Performance is analyzed using both the hard voting and soft voting criteria.
- For performance analysis, support vector machine (SVM), decision tree (DT), logistic regression (LR), and k nearest neighbor (KNN) models are used with three different feature extraction approaches including term frequency-inverse document frequency (TF-IDF), the bag of words (BoW), and hashing.
- For experiments, a large dataset of online meeting apps tweets has been collected. Dataset is labeled using the valence aware dictionary for sentiment reasoning (VADER) while for topic modeling, the latent Dirichlet allocation (LDA) approach is used. Performance is evaluated using accuracy, precision, recall, and F1 score. In addition, a comparison of the proposed model is carried out with the state-of-the-art approaches.

The rest of the paper is structured as follows: The 'Related Work' section discusses research works related to app reviews and hybrid approaches. The proposed research

methodology for app reviews sentiment analysis and its related contents are presented after that. It is followed by a discussion of the results. In the end, the conclusion is given in the last section.

## RELATED WORK

Reviews analysis has become one of the most widely researched areas over the past few years due to the popularity of social media platforms. In addition, many service providers provide online services and ask customers for feedback or views regarding the quality of services. Such reviews have significant importance to determine the quality of the services/products. However, it requires analyzing the text/views for user conceptions and perceptions. Especially negative sentiment reviews contain more important points for improving the quality. Keeping in view the importance of text analysis, a large body of work is available regarding sentiment analysis.

The study (*Rustam et al., 2020a*) investigates the Shopify app reviews using supervised machine learning models. The authors perform sentiment analysis for the Shopify app using the reviews dataset with a hybrid approach comprising logistic regression (LR), TF-IDF features, and chi-square (chi2) features. The Chi2 is used to select the important features for training while LR classifies the reviews into happy and unhappy and obtains a 79% accuracy score. Similarly, the authors use the word vector approach for app reviews sentiment analysis in *Fan et al. (2016)*. Experiments to show the effectiveness of vector-based features for sentiment analysis show that an 85.77% F1 score is obtained using Naive Bayes (NB). The study (*Rekanar et al., 2022*) performs sentiment analysis on an Irish health service executive's COVID-19 contact tracing app. Manual sentiment analysis on 1287 reviews extracted from Google and Apple play stores is performed.

Some studies also worked on employee reviews to evaluate employees' sentiments regarding the company's policies. For example, *Rustam et al. (2021a)* performs employee reviews classification using a supervised machine learning approach. The authors utilize multilayered perceptron (MLP) to achieve an 83% accuracy score. Review annotation plays a critical role in the performance of classification models and occasionally contradictions are found in the human and machine learning models annotation. The use of lexicon-based approaches has been investigated for data annotation and its impact on the models' performance (*Saad et al., 2021*). For example, study (*Trivedi & Singh, 2021*) uses the reviews regarding the online food delivery apps Swiggy, Zomato, and UberEats for sentiment analysis. The study shows the suitability of lexicon-based approaches for sentiment classification.

Investigating the suitability of features is an important aspect of sentiment analysis. Often, the change in the feature engineering method leads to a change in models' performance (*Khalid et al., 2020*; *Umer et al., 2021*). The study (*Rehan et al., 2022*) proposed an approach for employee reviews classification and evaluation. It uses an extra trees classifier (ETC) and bag of words (BoW) feature for employee reviews classification. The study uses both numerical and text features for employee reviews classification and achieved 100% and 79% accuracy scores, respectively. The study (*Tam,*

*Said & Tanriöver, 2021*), proposed a sentiment classification approach. They combined CNN and Bidirectional LSTM (Conv-BiLSTM) for tweets sentiment classification. Conv-bi-LSTM with Word2Vec performs significantly with 91.13% accuracy. Another study (*Jain, Saravanan & Pamula, 2021*), proposed a hybrid model CNN-LSTM for consumer sentiment analysis. They deployed the proposed model on qualitative user-generated content for sentiment analysis and achieved 91.3% accuracy.

Studies show that the performance of the ensemble and hybrid models is superior to that of single models for sentiment analysis (*Jamil et al., 2021*). For example, *Rupapara et al. (2021a)* uses a hybrid model of bi-LSTM models to obtain higher accuracy for sentiment classification. Similarly, *Rupapara et al. (2021b)* adopts a hybrid model of regression vector voting classifier for toxic sentiments classification. Keeping in view the performance of ensemble classifiers and voting mechanisms, this study adopts the voting approach for the proposed ensemble model. However, contrary to previous studies that use voting from different models, this study proposes the novel use of self-voting criteria for sentiment analysis of online meeting apps.

## PROPOSED APPROACH

This study utilizes a machine learning approach for the sentiment classification of online meeting app reviews. This analysis can help online meeting apps owner to improve the app quality to attract more users. We analyze the sentiments of users so that the app owners can get insights on important features of apps and improve them in light of users' sentiments.

The architecture of the proposed approach is shown in Fig. 1. For the proposed approach, initially, the dataset is collected from the Google Play store using the Google app reviews crawler. The collected dataset contains app reviews related to online meeting apps in their raw form and contains unnecessary and redundant information. To clean reviews text, several preprocessing steps are applied to reduce the complexity of the text. Afterward, the dataset is annotated using the lexicon-based technique VADER. For model training, feature extraction is performed. For this purpose, three feature extraction techniques are investigated including TF-IDF, BoW, and hashing. The performance of many machine learning models is analyzed including SVM, DT, LR, KNN, and RF. In the end, the models are evaluated in terms of accuracy, precision, recall, and F1 score. In addition to sentiment analysis, this study also performs topic modeling using the LDA model.

### Dataset description

The dataset is extracted from the Google Play store for several online meeting apps including 'Google Meet', 'Goto Meeting', 'Zoom Meeting', 'Skype', 'Hangouts', 'Microsoft Teams', and 'Webex Meeting'. These apps have been selected regarding their overall rating on the Google Play store. The app's reviews are extracted for the period of 12 October 2018 to 7 December 2021. This study considers only the reviews given in the English language. The reviews are collected using the Google Play scraper library. The collected dataset contains the review id, user name, the content of reviews, score by user for the app, thumbs up count, review created version, and data for the review posted. The reviews contain opinions of users regarding particular positive and negative features of an app. Besides criticism, such

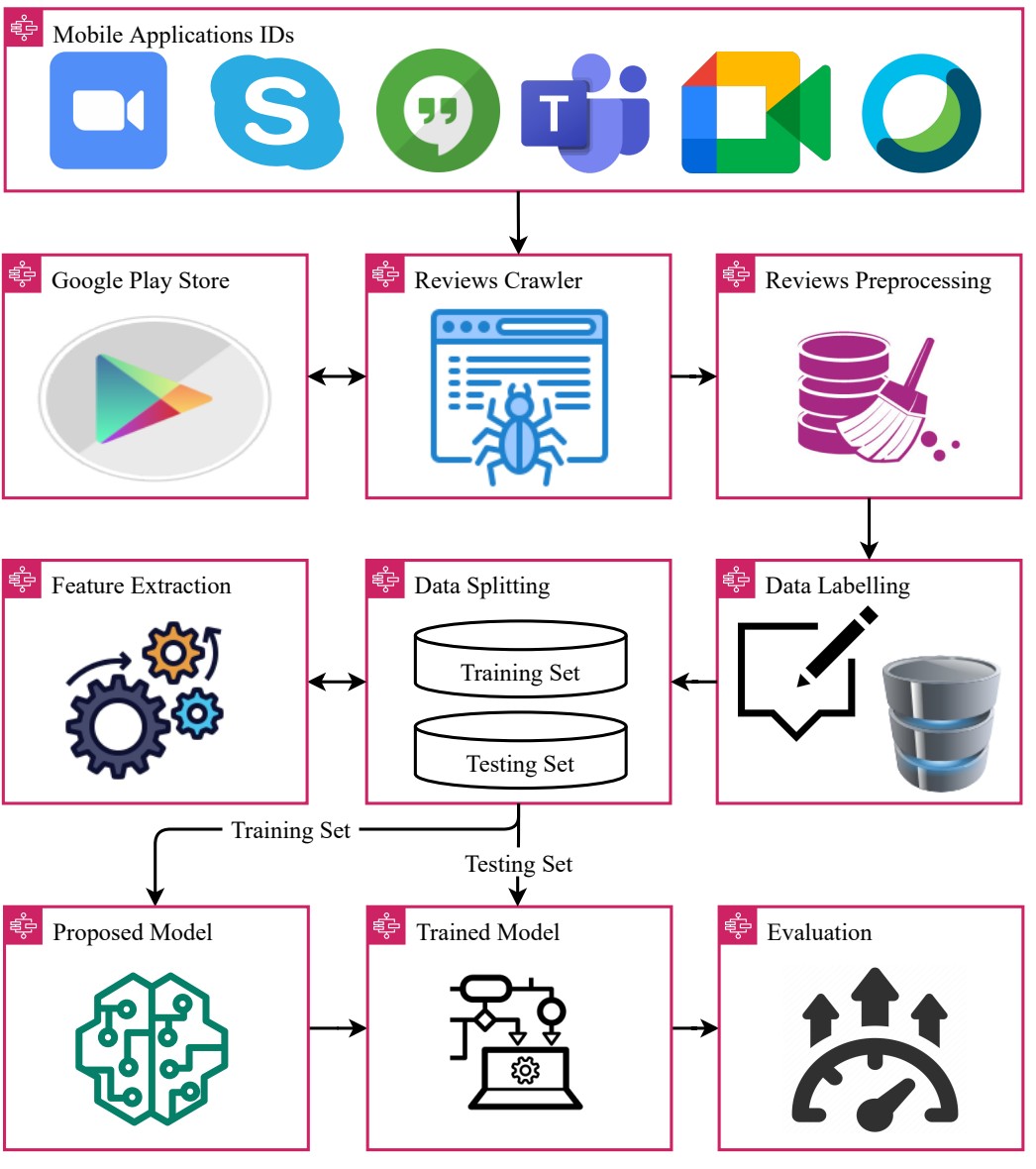

**Figure 1** Steps followed in the adopted methodology.

reviews also contain suggestions for improvement or the addition of new features. These reviews are very helpful for app companies to make changes according to user sentiments. Sample data from the collected dataset is shown in Table 1. The number of reviews varies for each app and the distribution of reviews is provided in Fig. 2.

## Preprocessing steps

Preprocessing is an important part of text analysis which helps to reduce the complexity of feature vectors and improves models' performance (*Mehmood et al., 2017*). The extracted dataset contains irrelevant and redundant information which can be removed to reduce the feature complexity without affecting the models' performance. Several preprocessing steps

| Table 1 | Dataset attributes and their description. | | |
|---------|-------------------------------------------|---|---|
| **No.** | **reviewId** | **userName** | **content** |
| 0 | gp:AOqp, …, | Rick S | Only works intermittently, …, |
| 1 | gp:AOq, …, | Angela Tudorii | I've been using Skype for, …, |
| 2 | gp:AOqp, … | Adriana Rodriguez | Horrible! Have not been ... |
| 3 | gp:AOqp, …, | Chloe | Took FOREVER to sign in, …, |
| **score** | **thumbsUpCount** | **reviewCreatedVersion** | **at** |
| 4 | 323 | 8.78.0.164 | 11/14/2021 6:42 |
| 2 | 238 | 8.78.0.164 | 11/14/2021 7:17 |
| 4 | 64 | 8.78.0.164 | 11/28/2021 22:34 |
| 1 | 33 | 8.78.0.164 | 11/25/2021 7:15 |

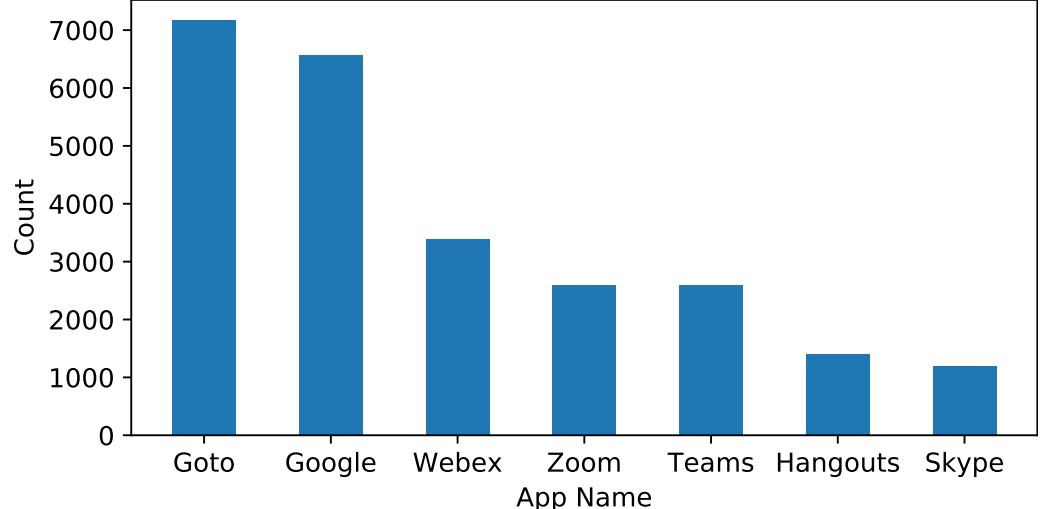

**Figure 2** Distribution of reviews for each app.

are used to clean data such as removal of numbers, removal of punctuation, conversion to lowercase, stemming, and removal of stopwords.

- **Removal of numbers:** Occasionally user reviews contain numbers that do not contribute to sentiment classification. These numbers are removed using the Python function isalpha() which ensures that only characters are forwarded for further preprocessing.
- **Removal of punctuation:** Text contain lots of punctuation marks that help humans understand the intended meaning. However, punctuation is not useful for sentiment analysis using machine learning models. The punctuation marks are removed to reduce feature complexity.
- **Convert to lowercase:** This preprocessing step helps to reduce the complexity of the feature vector. Feature extraction techniques consider lower and upper case words as unique words. For example, 'User', 'user', and 'USER' convey the same meaning for

humans but feature extraction techniques treat them as unique words. Conversion to lowercase helps to reduce complexity.

- **Stemming:** Stemming is another very helpful preprocessing step to reduce the feature complexity. It changes different forms of the same word to its root form. For example, 'go', 'going' and 'goes' are changed to their basic form 'go'. Porter stemmer library is used for this purpose.
- **Removal of stopwords:** Text contain lots of stopwords to improve text readability for humans, for machine learning approaches, they are useless. Consequently, removing the words such as 'is', 'an', 'the', 'and' etc. helps to reduce the feature set and improve classification performance.

Sample text data from the collected dataset, before and after the preprocessing steps is shown in Table 2.

## Valence aware dictionary for sentiment reasoning

VADER is used for sentiment extraction from text data. VADER analyzes the polarity and sensitivity of sentiment in the text and finds the sentiment score by adding the intensities of each word in the text (*Hutto & Gilbert, 2014*). The sentiment score range varies between $-4.0$ to $+4.0$, where $-4$ is the most negative and $+4$ is the most positive sentiment score. The midpoint 0 represents a neutral sentiment. Figure 3 shows the ratio of positive, negative, and neutral sentiments in the dataset extracted using VADER.

## Latent dirichlet allocation

LDA is a modeling technique used to extract topics from a text corpus. Latent means 'hidden' which shows that it is used to extract hidden topics in data (*Blei, Ng & Jordan, 2003*). LDA is based on Dirichlet distributions and processes and uses two metrics for topic modeling. Probability distribution of topics in documents and probability distribution of words in topics are used for topic modeling (*LDA, 2018*).

## Feature engineering

The feature extraction techniques are required for training the machine learning models. This study uses three feature extraction techniques to train the models.

### *Bag of words*

The bag of words (BoW) is the simplest technique used for feature extraction from text data (*Rustam et al., 2021a*). The BoW technique counts the appearance of each unique term from the corpus and makes a vector for the machine learning models. Depending upon the number of occurrences of different words, text similarity can be determined using the BoW feature vector. BoW features are extracted using the CountVectorizer Sci-Kit learn library.

### *Term frequency-inverse document frequency*

TF-IDF is a widely used feature selection technique in text classification domain (*Rustam et al., 2020a*). Contrary to simple frequency count in BoW, TF-IDF makes a weighted feature. TF counts the frequency while IDF calculates the weights of each term in the corpus. IDF

**Table 2  Preprocessing results on sample reviews.**

| Reviews | After preprocessing |
|---|---|
| I would prefer to see the app show any video calls in a minimized window on movile devices like it would in the past. | prefer see app show any video call minimizi window movile devic past |
| I think they're actively trying to make it worse. | think activi try make worse |

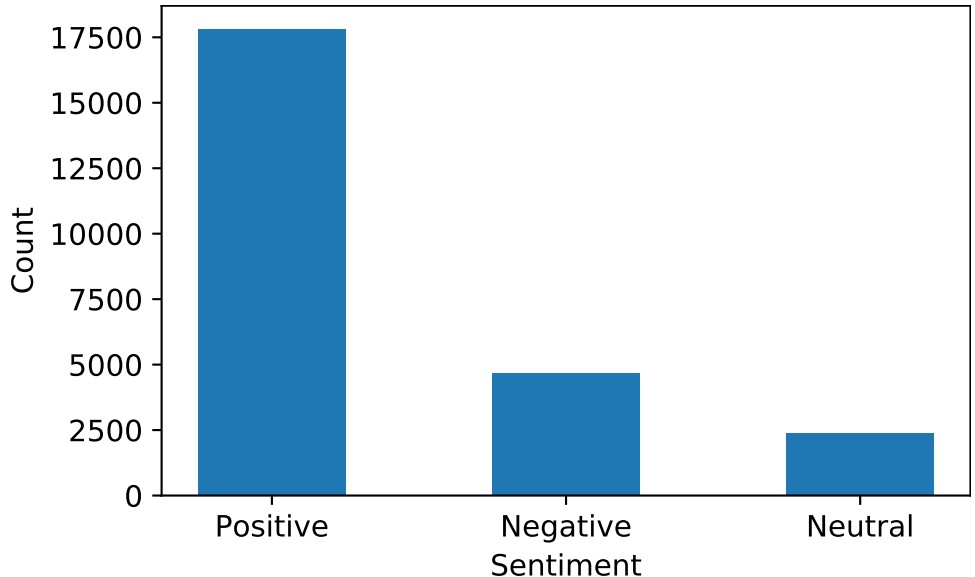

**Figure 3  Distribution of sentiments for the collected dataset.**

considers less frequent words more important and assigns them higher weights. TF, IDF, and TF-IDF are calculated using

$$tf = TF_{p,q} \tag{1}$$

where $tf$ is the term frequency of term $p$ in document $q$.

$$idf = log\frac{N_r}{D_p} \tag{2}$$

where $N_r$ is the number of documents in a corpus and $D_p$ is the number of documents containing the term $p$. TF-IDF can be obtained by multiplying $tf$ and $idf$.

### Hashing

Hashing is another text feature extraction technique that converts text corpus into a matrix of token occurrences (*Kulkarni & Shivananda, 2019*). It is a memory-efficient algorithm that requires low memory for a large dataset. It does not store a vocabulary dictionary in memory and is very suitable for large datasets.

**Table 3   Optimized hyperparameters setting for machine learning models.**

| Model | Hyper-parameters | Tuning range |
|---|---|---|
| DT | max_depth = 300 | max_depth = {2 to 500} |
| SVM | kernel = 'linear', $C = 1.0$ | kernel = {'linear', 'poly', 'sigmoid'}, C = {1.0 to 5.0} |
| LR | Solver = saga, $C = 1.0$, multi_class = multinomial | Solver = {saga, sag, liblinear}, C = {1.0 to 5.0}, multi_class = {ovr, multinomial} |
| KNN | n_neighbors = 5 | n_neighbors = {2 to 8} |

## Machine learning models

This study uses four machine learning models including SVM, DT, LR, and KNN to validate the proposed self-voting approach. These models are used with their best hyperparameters setting according to the dataset. To select the best hyperparameters values ranges are obtained from the literature and fine-tuned to obtain the best performance (*Rupapara et al., 2021a*; *Mujahid et al., 2021*). The hyperparameter setting and tuning range are given in Table 3.

## Self-Voting classifier

This study proposes a novel voting classifier, called a self-voting classifier. Traditional ensemble models follow a group voting mechanism, using heterogeneous models where the output of multiple models is combined using soft or hard voting criteria. Since the performance of different models varies, combining the prediction of multiple models improves the classification performance (*Rustam et al., 2020b*; *Rustam et al., 2019*; *Rupapara et al., 2021b*). Contrary to the group voting from heterogeneous models, this study adopts the self-voting ensemble where the output of the three different variants of SVM is combined to make the final prediction. Since the performance of a model varies concerning the features fed for training, the idea is to feed multiple features to the same model and combine them to make the ensemble. Three SVM variants have been trained on different feature vectors including BoW, TF-IDF, and hashing features. The performance of the self-voting approach is investigated both using the soft and hard voting criteria.

Figure 4 shows the process followed for soft voting (SV) where the probabilities predicted from each SVM variant is considered to calculate the average prediction probability of each class. The SVM-SV approach follows these steps. First, TF-IDF features are used for training the SVM using Eq. (3).

$$tfidf = tf_{p,q} * log\left(\frac{N_r}{D_q}\right)$$ 

(3)

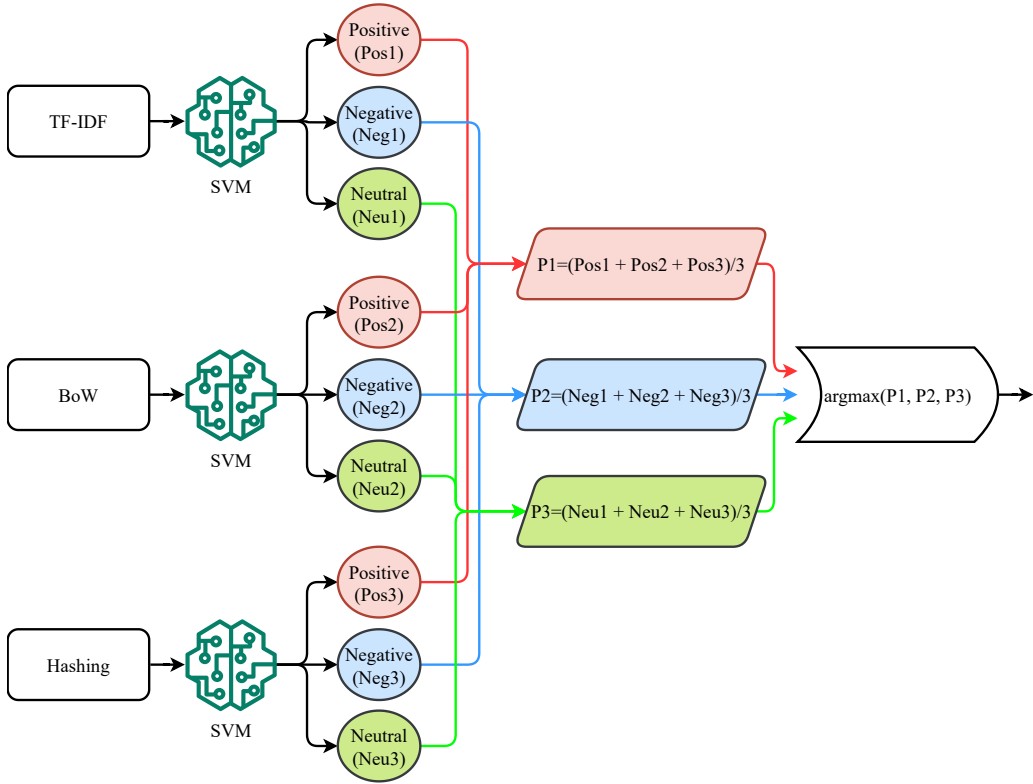

**Figure 4 Soft voting mechanism used for the proposed approach.**

where *tfidf* gives weights for terms in the corpus using the TF-IDF.

$$tfidf_{set} = \begin{pmatrix} F_1 & F_2 & ... & F_m \\ & & & \\ tfidf_{1x1} & tfidf_{1x2} & ... & tfidf_{1xm} \\ tfidf_{2x1} & tfidf_{2x2} & ... & tfidf_{2xm} \\ . & . & & . \\ . & . & & . \\ . & . & & . \\ tfidf_{nx1} & tfidf_{nx2} & ... & tfidf_{nxm} \end{pmatrix}.$$ (4)

The $tfidf_{set}$ is a feature set extracted using the TF-IDF technique and *m* is the number of features.

The unique words that belong to $(N_r)$ number of reviews can be represented as

$$f_1, f_2, ..., f_n \epsilon N_r \ and \ N_= n.$$ (5)

Similar to TF-IDF, two SVM variants are trained on BoW and hashing features, respectively.

$$bow = Count(t, N_{r,i})$$ (6)

where BoW is the count of term $t$ in a review $N_{(r,i)}$ where $N_{(r,i)} \in N_r$ and below $bow_{set}$ is a feature set extracted using the BoW technique.

$$bow_{set} = \begin{pmatrix} F_1 & F_2 & \dots & F_m \\ bow_{1x1} & bow_{1x2} & \dots & bow_{1xm} \\ bow_{2x1} & bow_{2x2} & \dots & bow_{2xm} \\ . & . & & . \\ . & . & & . \\ . & . & & . \\ bow_{nx1} & bow_{nx2} & \dots & bow_{nxm} \end{pmatrix} \qquad (7)$$

For hashing features, the feature set can be defined as

$$h = hash(str) = str[0] + str[1]pn^1 + \dots + str[n]pn^n \qquad (8)$$

where $h$ is the value of a string ($str$) calculated using hashing vectorizer function, $pn$ is a prime number, $str[i]$ is a character code, $q$ is the index value and $p$ is the value for the number of $str$ strings.

$$hash_{set} = \begin{pmatrix} F_1 & F_2 & \dots & F_m \\ h_{1x1} & h_{1x2} & \dots & h_{1xm} \\ h_{2x1} & h_{2x2} & \dots & h_{2xm} \\ . & . & & . \\ . & . & & . \\ . & . & & . \\ h_{nx1} & h_{nx2} & \dots & h_{nxm} \end{pmatrix}. \qquad (9)$$

Using the $tfidf_{set}$, $bow_{set}$, and $hash_{set}$ feature sets, three SVM variants are trained as follows

$$svm_{t1} = SVM(tfidf_{set}) \qquad (10)$$

$$svm_{t2} = SVM(bow_{set}) \qquad (11)$$

$$svm_{t3} = SVM(hash_{set}) \qquad (12)$$

where $svm_{t1}$, $svm_{t2}$, and $svm_{t3}$ are trained SVM using each feature set and can be combined to make the final prediction using SV criteria.

$$pos_{p1}, neg_{p1}, neu_{p1} = svm_{t1}(TD_{features}) \qquad (13)$$

$$pos_{p2}, neg_{p2}, neu_{p2} = svm_{t2}(TD_{features}) \qquad (14)$$

$$pos_{p3}, neg_{p3}, neu_{p3} = svm_{t3}(TD_{features}) \tag{15}$$

where $pos_p$, $neg_p$, and $neu_p$ are probabilities for positive, negative, and neutral target classes, respectively and $TD_{features}$ are features for test samples.

$$p1 = \frac{pos_{p1} + pos_{p1} + pos_{p1}}{3} \tag{16}$$

$$p2 = \frac{pos_{p2} + pos_{p2} + pos_{p2}}{3} \tag{17}$$

$$p3 = \frac{pos_{p3} + pos_{p3} + pos_{p3}}{3} \tag{18}$$

where $p1$, $p2$, and $p3$ are probabilities for positive, negative, and neutral classes using TF-IDF, BoW, and Hashing features, respectively. SVM-SV uses the *argmax* function in the end to find the class with the highest probability.

$$final\ prediction = argmax\{p1, p2, p3\}. \tag{19}$$

For hard voting (HV), the predicted class from each SVM variant is considered for the final prediction, as shown in Fig. 5. SVM-HV method uses majority voting criteria to make the final prediction. Each SVM variant predicts a target class (positive, negative, or neutral) using each feature set and then the SVM-HV performs voting on the predicted class. In case of a tie in voting, a higher weight is awarded to the minority class in the dataset which is the neutral class for this dataset.

$$p1 = SVM(tfidf_{set}) \tag{20}$$

$$p2 = SVM(bow_{set}) \tag{21}$$

$$p3 = SVM(hash_{set}) \tag{22}$$

where $p1$, $p2$, and $p3$ are predictions by SVM variants with different feature sets. The majority voting function is used on these predictions to make the final prediction. In the case of tie *final prediction* $\in$ *minority class*.

$$final\ prediction = mode\{p1, p2, p3\}. \tag{23}$$

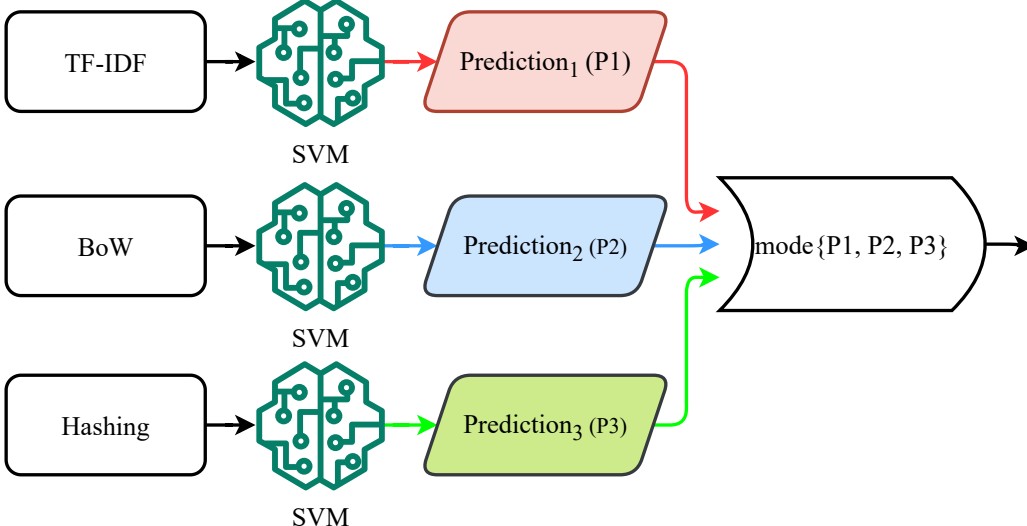

**Figure 5** **Hard voting mechanism used for the proposed approach.**

---

**Algorithm 1** Proposed SVM-SV algorithm

    **Input:** Apps Reviews

    **Output:** *Positive|Negative|Neutral*

1:  Def Model_Training():

2:     $SVM_T \leftarrow$ SVM(TF-IDF_Features)

3:     $SVM_B \leftarrow$ SVM(BoW_Features)

4:     $SVM_H \leftarrow$ SVM(Hashing_Features)

5:  **for** $i$ in Test_Corpus **do**

6:     $P1 \leftarrow SVM_T(i)$

7:     $P2 \leftarrow SVM_B(i)$

8:     $P3 \leftarrow SVM_H(i)$

9:     $SVM - SV(Pred) \leftarrow mode\{P1, P2, P3\}$

10:  **end for**

11:  *Positive|Negative|Neutral* $\leftarrow SVM - SV$ prediction

---

Algorithm 1 shows the steps of the proposed SVM-SV model. Three different variants of SVM are trained as shown in lines 2 to 4 of Algorithm 1, where $SVM_T$ indicates the SVM model trained using TF-IDF features, $SVM_B$ is the model trained with BoW features while $SVM_H$ is the model trained using Hashing features. Lines 6 to 8 show the predictions made from each model where $P1$, $P2$, and $P3$ are the prediction by trained $SVM_T$, $SVM_B$ and $SVM_H$, respectively. In the end, *mode* of $P1$, $P2$, and $P3$ are taken to predict the final label of the test sample. It is the soft voting criterion; it infers that if two models out of three predict the test sample as positive, the final prediction will be positive.

Algorithm 2 shows the steps of the proposed SVM-HV model. Its input is app reviews while the output is the label of the sentiments for particular reviews. Three SVM are trained using each feature extraction method where $SVM_T$, $SVM_B$, and $SVM_H$ indicate the models trained using TF-IDF, BoW, and Hashing features, respectively. In this algorithm, the probability of each sentiment is taken from each trained model, where $Pos_i$, $Neg_i$, and $Neu_i$ are probabilities by each model for positive, negative, and neutral target classes, respectively, as given in lines 6 to 14 of Algorithm 2. $Prob\_Pos$, $Prob\_Neg$, and $Prob\_Neu$ are the average probabilities that are calculated using all models' probabilities. Average probability is calculated by taking the summation of the probability of positive class from each model and dividing it by 3. The same procedure is adopted for negative and neutral sentiments. In the end, the *argmax* is taken to predict the final label for the test sample.

---

**Algorithm 2** Proposed SVM-HV algorithm

---

**Input:** Apps Reviews

**Output:** *Positive|Negative|Neutral*

1: Def Model_Training():
2:     $SVM_T \leftarrow$ SVM(TF-IDF_Features)
3:     $SVM_B \leftarrow$ SVM(BoW_Features)
4:     $SVM_H \leftarrow$ SVM(Hashing_Features)
5: **for** $i$ in Test_Corpus **do**
6:     $Pos1 \leftarrow SVM_T(i)$
7:     $Neg1 \leftarrow SVM_T(i)$
8:     $Neu1 \leftarrow SVM_T(i)$
9:     $Pos2 \leftarrow SVM_B(i)$
10:    $Neg2 \leftarrow SVM_B(i)$
11:    $Neu2 \leftarrow SVM_B(i)$
12:    $Pos3 \leftarrow SVM_H(i)$
13:    $Neg3 \leftarrow SVM_H(i)$
14:    $Neu3 \leftarrow SVM_H(i)$
15:    $Prob\_Pos \leftarrow \frac{(Pos1+Pos2+Pos3)}{3}$
16:    $Prob\_Neg \leftarrow \frac{(Neg1+Neg2+Neg3)}{3}$
17:    $Prob\_Neu \leftarrow \frac{(Neu1+Neu2+Neu3)}{3}$
18:    $SVM-HV(Pred) \leftarrow argmax\{Prob\_Pos, Prob\_Neg, Prob\_Neu\}$
19: **end for**
20: *Positive|Negative|Neutral* $\leftarrow SVM-HV$ prediction

---

# RESULTS AND DISCUSSION

This section presents and discusses the performance of machine learning models for app reviews sentiment analysis. The performance of the proposed SVC-SV and SVC-HV is evaluated in terms of accuracy, precision, recall, and F1 score.

**Table 4   Number of records for training and testing datasets.**

| Target | Training set | Testing set | Total |
|---|---|---|---|
| Positive | 14,224 | 3,592 | 17,816 |
| Negative | 3,727 | 943 | 4,670 |
| Neutral | 1,949 | 440 | 2,389 |
| Total | 19,900 | 4,975 | 24,875 |

## Experimental setup

For experiments, this study used an Intel Core i7 11th generation machine with the Windows operating system. To implement the proposed approach, Jupyter notebook is used with the Python language and Sci-kit learn, TensorFlow, NLTK, and Pandas libraries are used. Data splitting is done for model training and testing in ratios of 80% and 20%, respectively. The dataset contains three target classes including positive, negative, and neutral. The number of samples in the dataset after the data split is given in Table 4.

## Results for sentiment classification

Table 5 shows the results of SVM with BoW, TF-IDF, and hashing features. It also contains the results of proposed approaches SVC-SV and SVC-HV. SVM performs significantly better with TF-IDF and hashing features and obtained a 0.98 accuracy score with each approach. On the other hand, BoW features do not show good results and SVM has a 0.95 accuracy score. The performance with TF-IDF and hashing features is more significant because of the significant feature sets generated by these techniques. TF-IDF assigns weight to each feature and shows better results as compared to simple term count from the BoW technique. Similarly, hashing generates a less complex feature set for model training which helps to increase models' performance. SVC-SV is also good, similar to other features with SVM, however, SVC under hard voting under majority voting criteria outperforms all other approaches with a 1.00 accuracy score. This significant performance is primarily based on the combination of multiple variants of SVM trained on different features. It can be observed that different SVM variants show different per class accuracy for positive, negative, and neutral classes. For example, SVM with TF-IDF is good for the neutral class while using hashing feature is good to obtain the best performance for the positive class. Combining these variants trained on different features helps to obtain the best performance in all the classes as the SVM variants complement each other.

The self-voting approach has been validated using several machine learning models including DT, KNN, and LR. Table 6 shows the results using the DT model in terms of accuracy, precision, recall, and F1 score. Other than the self-voting approach, DT shows the best result when used with BoW features and obtains a 0.87 accuracy score as compared to TF-IDF and hashing features. DT is a simple rule-based model and can perform better using a simple feature set such as extracted by the BoW. DT with TF-IDF and hashing has marginally low performance with a 0.86 accuracy score for each feature set. The best performance is obtained when it is used with SVC-HV with a 0.88 accuracy score. Besides accuracy, precision, recall, and F1 score values are also superior to that of other features.

**Table 5   Results using different feature engineering approaches with SVM.**

| Model | Accuracy | Target | Precision | Recall | F1 Score |
|---|---|---|---|---|---|
| BoW | 0.95 | Negative | 0.90 | 0.90 | 0.90 |
| | | Neutral | 0.85 | 0.93 | 0.89 |
| | | Positive | 0.98 | 0.97 | 0.98 |
| | | Avg. | 0.91 | 0.94 | 0.92 |
| TF-IDF | 0.98 | Negative | 0.98 | 0.97 | 0.97 |
| | | Neutral | 0.96 | 0.96 | 0.96 |
| | | Positive | 0.99 | 0.99 | 0.99 |
| | | Avg. | 0.98 | 0.97 | 0.97 |
| Hashing | 0.98 | Negative | 0.97 | 0.93 | 0.95 |
| | | Neutral | 0.90 | 0.96 | 0.93 |
| | | Positive | 0.99 | 0.99 | 0.99 |
| | | Avg. | 0.95 | 0.96 | 0.96 |
| SVC-SV using SVM | 0.98 | Negative | 0.99 | 0.93 | 0.96 |
| | | Neutral | 0.96 | 0.95 | 0.95 |
| | | Positive | 0.98 | 1.00 | 0.99 |
| | | Avg. | 0.98 | 0.96 | 0.97 |
| SVC-HV using SVM | 1.00 | Negative | 1.00 | 1.00 | 1.00 |
| | | Neutral | 1.00 | 1.00 | 1.00 |
| | | Positive | 1.00 | 1.00 | 1.00 |
| | | Avg. | 1.00 | 1.00 | 1.00 |

Table 7 shows the performance results of the LR model using BoW, TF-IDF, hashing features, and the SVC approach. LR shows better performance as compared to DT, however, its performance is inferior to SVM. LR performance with the SVC approach is more significant as compared to an individual feature but SVC-SV achieved a 0.95 accuracy score which is the highest as compared to results using other features.

KNN is another model that is used for experiments deployed with the proposed SVC approach. Experimental results given in Table 8 indicate that the proposed approach shows significant improvements over other approaches. On average, the performance of KNN is not good as compared to SVM, DT, and LR as it has accuracy scores of 0.75, 0.76, and 0.76 when used with BoW, TF-IDF, and hashing features, respectively. KNN tends to show poor performance with large datasets as compared to linear models such as SVM and LR which are more suitable for large feature sets, such as the dataset used in this study. Using the proposed SVC approach, the accuracy score of KNN is improved to 0.78 from 0.76.

## Performance of deep learning models on apps reviews dataset

In comparison with our proposed approach using the machine learning models, this study also deploys some state of the arts deep learning models. For this purpose, long short-term memory (LSTM) (*Rupapara et al., 2021a*), gated recurrent unit(GRU) (*Dey & Salem, 2017*), convolutional neural networks(CNN) (*Luan & Lin, 2019*), and recurrent neural networks (RNN) are used. The architecture of these models is presented in Table 9.

**Table 6 Performance of DT with different feature engineering approaches.**

| Model | Accuracy | Target | Precision | Recall | F1 Score |
|---|---|---|---|---|---|
| BoW | 0.87 | Negative | 0.74 | 0.69 | 0.71 |
| | | Neutral | 0.72 | 0.82 | 0.77 |
| | | Positive | 0.93 | 0.93 | 0.93 |
| | | Avg. | 0.79 | 0.81 | 0.80 |
| TF-IDF | 0.86 | Negative | 0.72 | 0.68 | 0.70 |
| | | Neutral | 0.69 | 0.77 | 0.73 |
| | | Positive | 0.92 | 0.92 | 0.92 |
| | | Avg. | 0.78 | 0.79 | 0.78 |
| Hashing | 0.86 | Negative | 0.72 | 0.68 | 0.70 |
| | | Neutral | 0.69 | 0.77 | 0.73 |
| | | Positive | 0.92 | 0.92 | 0.92 |
| | | Avg. | 0.78 | 0.79 | 0.78 |
| SVC-SV using DT | 0.85 | Negative | 0.65 | 0.70 | 0.67 |
| | | Neutral | 0.69 | 0.72 | 0.71 |
| | | Positive | 0.92 | 0.90 | 0.91 |
| | | Avg. | 0.76 | 0.77 | 0.76 |
| SVC-HV using DT | 0.88 | Negative | 0.74 | 0.70 | 0.72 |
| | | Neutral | 0.74 | 0.80 | 0.77 |
| | | Positive | 0.93 | 0.93 | 0.93 |
| | | Avg. | 0.80 | 0.81 | 0.80 |

The models use dropout layers, dense layers, and embedding layers as common among all models. The dropout layer is used to reduce the probability of model over-fitting and reduces the complexity of model learning by dropping neurons randomly. The embedding layer takes input and converts each word in reviews into vector form for model training. The dense layer is used with three neurons and a Softmax activation function to generate the desired output. Models are compiled with categorical cross-entropy function because of multi-class data and 'adam' optimizer is used for parameters optimization (*Zhang, 2018*). In the end, all models are fitted with 100 epochs and a batch size of 64.

Experimental results using deep learning models are given in Table 10. Results show that LSTM and GRU outperform other deep learning models with 0.92 and 0.91 accuracy scores, respectively. The performance of LSTM and GRU shows that the recurrent architecture model shows significantly better performance than other models on text data. RNN is also better compared to CNN which has the lowest accuracy of 0.81. The mechanism of eliminating unused information and storing the sequence of information makes recurrent applications a strong tool for text classification tasks. On the other hand, CNN requires a large feature set to perform better which in the case of this study does not seem so.

## Comparison with other studies

The performance of the proposed approach is compared with other recent studies on sentiment analysis. In this regard, the state-of-the-art models from previous studies are deployed on the current dataset and the results are compared. First, the study (*Rustam et*

**Table 7   Performance of DT using different feature engineering approaches.**

| Model | Accuracy | Target | Precision | Recall | F1 Score |
|---|---|---|---|---|---|
| BoW | 0.94 | Negative | 0.92 | 0.84 | 0.88 |
| | | Neutral | 0.81 | 0.78 | 0.80 |
| | | Positive | 0.96 | 0.98 | 0.97 |
| | | Avg. | 0.90 | 0.87 | 0.88 |
| TF-IDF | 0.94 | Negative | 0.95 | 0.86 | 0.90 |
| | | Neutral | 0.92 | 0.72 | 0.80 |
| | | Positive | 0.94 | 0.99 | 0.97 |
| | | Avg. | 0.94 | 0.86 | 0.89 |
| Hashing | 0.94 | Negative | 0.94 | 0.81 | 0.87 |
| | | Neutral | 0.85 | 0.79 | 0.82 |
| | | Positive | 0.95 | 0.99 | 0.97 |
| | | Avg. | 0.91 | 0.86 | 0.89 |
| SVC-SV using LR | 0.95 | Negative | 0.94 | 0.85 | 0.89 |
| | | Neutral | 0.87 | 0.79 | 0.83 |
| | | Positive | 0.95 | 0.99 | 0.97 |
| | | Avg. | 0.92 | 0.88 | 0.90 |
| SVC-HV using LR | 0.94 | Negative | 0.94 | 0.84 | 0.89 |
| | | Neutral | 0.87 | 0.77 | 0.82 |
| | | Positive | 0.95 | 0.99 | 0.97 |
| | | Avg. | 0.92 | 0.87 | 0.89 |

al., 2019) used an ensemble model which is the combination of LR and stochastic gradient descent classifier (SGDC) for sentiment classification. The ensemble model is deployed on the current dataset and it obtained a 0.90 accuracy score. The study (Rustam et al., 2021b) used a hybrid approach for sentiment classification related to COVID-19 tweets. The study used an extra tree classifier and feature union technique for sentiment classification. The study (Rustam et al., 2020a) used a hybrid approach which is a combination of TF-IDF features, Chi-square feature selection technique, and LR model. The study (Tam, Said & Tanriöver, 2021) proposed a hybrid model ConvBiLSTM using CNN and BiLSTM networks for tweets sentiment classification and similarly, another study (Jain, Saravanan & Pamula, 2021) proposed a hybrid model CNN-LSTM for sent for consumer sentiment analysis. Performance comparison results of these studies are provided in Table 11.

## Statistically significant T-test

A statistical $T$-test is performed to show the significance of the proposed approach. $T$-test accepts the null hypothesis if the compared values are statistically the same and reject the null hypothesis if the compared values are statistically different (Omar et al., 2021). We deploy the $T$-test on the models' performance with each feature and the proposed self-voting. We evaluate performance in terms of T-statistic and critical value (CV). The T-statistic value is greater than the CV in all cases which means that for all cases the null hypothesis is rejected. T-statistic results are shown in Table 12. These results show that all cases are statistically different in comparison with the proposed approach.

**Table 8  Performance of KNN with SVC and different features.**

| Model | Accuracy | Target | Precision | Recall | F1 Score |
|-------|----------|--------|-----------|--------|----------|
| BoW | 0.75 | Negative | 0.70 | 0.33 | 0.45 |
| | | Neutral | 0.33 | 0.64 | 0.43 |
| | | Positive | 0.86 | 0.87 | 0.86 |
| | | Avg. | 0.63 | 0.61 | 0.58 |
| TF-IDF | 0.76 | Negative | 0.65 | 0.42 | 0.51 |
| | | Neutral | 0.32 | 0.37 | 0.34 |
| | | Positive | 0.83 | 0.90 | 0.86 |
| | | Avg. | 0.60 | 0.56 | 0.57 |
| Hashing | 0.76 | Negative | 0.65 | 0.40 | 0.50 |
| | | Neutral | 0.39 | 0.40 | 0.39 |
| | | Positive | 0.84 | 0.92 | 0.88 |
| | | Avg. | 0.63 | 0.57 | 0.59 |
| SVC-SV using KNN | 0.78 | Negative | 0.77 | 0.34 | 0.47 |
| | | Neutral | 0.41 | 0.45 | 0.43 |
| | | Positive | 0.82 | 0.93 | 0.87 |
| | | Avg. | 0.67 | 0.57 | 0.59 |
| SVC-HV using KNN | 0.78 | Negative | 0.68 | 0.41 | 0.51 |
| | | Neutral | 0.39 | 0.44 | 0.41 |
| | | Positive | 0.84 | 0.92 | 0.88 |
| | | Avg. | 0.64 | 0.59 | 0.60 |

**Table 9  Architecture of deep learning models used for experiments.**

| LSTM | GRU |
|------|-----|
| Embedding(5000,100, input_length) | Embedding(5000,100, input_length) |
| Dropout(0.2) | Dropout(0.2) |
| LSTM(128) | GRU(128) |
| Dropout(0.2) | Dense(16) |
| Dense(3, activation='softmax') | Dense(3, activation='softmax') |
| **CNN** | **RNN** |
| Embedding(5000,100, input_length) | Embedding(5000,100, input_length) |
| Conv1D(128, 4, activation='relu') | Dropout(0.2) |
| MaxPooling1D(pool_size=4) | SimpleRNN(100) |
| Flatten() | Dense(16) |
| Dense(16) | Dense(3, activation='softmax') |
| Dense(3, activation='softmax') | |

**Notes.**
loss='categorical_crossentropy', optimizer='adam', epochs=100.

## LDA topic extraction and topic sentiment visualization

This study also carried out topic modeling using the LDA approach. The topics are extracted from all app reviews, as well as, each app review to show the topic-wise users' sentiments. We used the LDA model to extract the top four topics from review data.

For topic modeling, the LDA is used with three hyperparameters including n_components, random_state, and evaluate_every. The n_components parameter is used

**Table 10   Performance comparison of deep learning models.**

| Model | Accuracy | Target | Precision | Recall | F1 Score |
|---|---|---|---|---|---|
| LSTM | 0.92 | Negative | 0.83 | 0.83 | 0.83 |
| | | Neutral | 0.81 | 0.76 | 0.79 |
| | | Positive | 0.95 | 0.96 | 0.96 |
| | | Avg. | 0.87 | 0.85 | 0.86 |
| GRU | 0.91 | Negative | 0.82 | 0.79 | 0.81 |
| | | Neutral | 0.81 | 0.73 | 0.77 |
| | | Positive | 0.94 | 0.96 | 0.95 |
| | | Avg. | 0.86 | 0.83 | 0.84 |
| CNN | 0.81 | Negative | 0.67 | 0.68 | 0.67 |
| | | Neutral | 0.52 | 0.38 | 0.44 |
| | | Positive | 0.87 | 0.90 | 0.89 |
| | | Avg. | 0.69 | 0.65 | 0.67 |
| RNN | 0.87 | Negative | 0.73 | 0.75 | 0.74 |
| | | Neutral | 0.77 | 0.70 | 0.73 |
| | | Positive | 0.93 | 0.93 | 0.93 |
| | | Avg. | 0.81 | 0.79 | 0.80 |

**Table 11   Comparative analysis of performance with other approaches.**

| Ref | Year | Approach | Accuracy | Precision | Recall | F1 Score |
|---|---|---|---|---|---|---|
| *Rustam et al. (2020a)* | 2021 | LR + Chi2 | 0.91 | 0.89 | 0.80 | 0.84 |
| *Tam, Said & Tanriöver (2021)* | 2021 | ConvBiLSTM | 0.82 | 0.72 | 0.64 | 0.67 |
| *Jain, Saravanan & Pamula (2021)* | 2021 | CNN-LSTM | 0.82 | 0.71 | 0.66 | 0.68 |
| *Rustam et al. (2019)* | 2019 | LR+SGDC Model TF-IDF Features | 0.90 | 0.83 | 0.82 | 0.82 |
| *Rustam et al. (2021b)* | 2021 | ETC Model(TF-IDF + BoW) FU | 0.83 | 0.86 | 0.57 | 0.63 |
| Curent study | 2021 | SVM + SVM + SVM (HV) and TF-IDF + BoW + Hashing Features | 1.00 | 1.00 | 1.00 | 1.00 |
| | 2021 | SVM + SVM + SVM (SV) and TF-IDF + BoW + Hashing Features | 0.98 | 0.98 | 0.96 | 0.97 |

**Table 12   *T*-test evaluation values.**

| Techniques | T-statistic | CV | Null hypothesis |
|---|---|---|---|
| BoW Vs HV | 2.038 | 0 | reject |
| BoW Vs SV | 1.188 | 0 | reject |
| TF-IDF Vs HV | 3.000 | 0 | reject |
| TF-IDF Vs SV | 0.775 | 0 | reject |
| Hashing Vs HV | 3.000 | 0 | reject |
| Hashing Vs SV | 0.775 | 0 | reject |

with value four indicating that four topics will be extracted with this setting; random_state with value is 10, and evaluate_every value is −1. The most commonly discussed topics are 'easy use', 'join meeting', 'online class', and 'virtual background'. We illustrate these

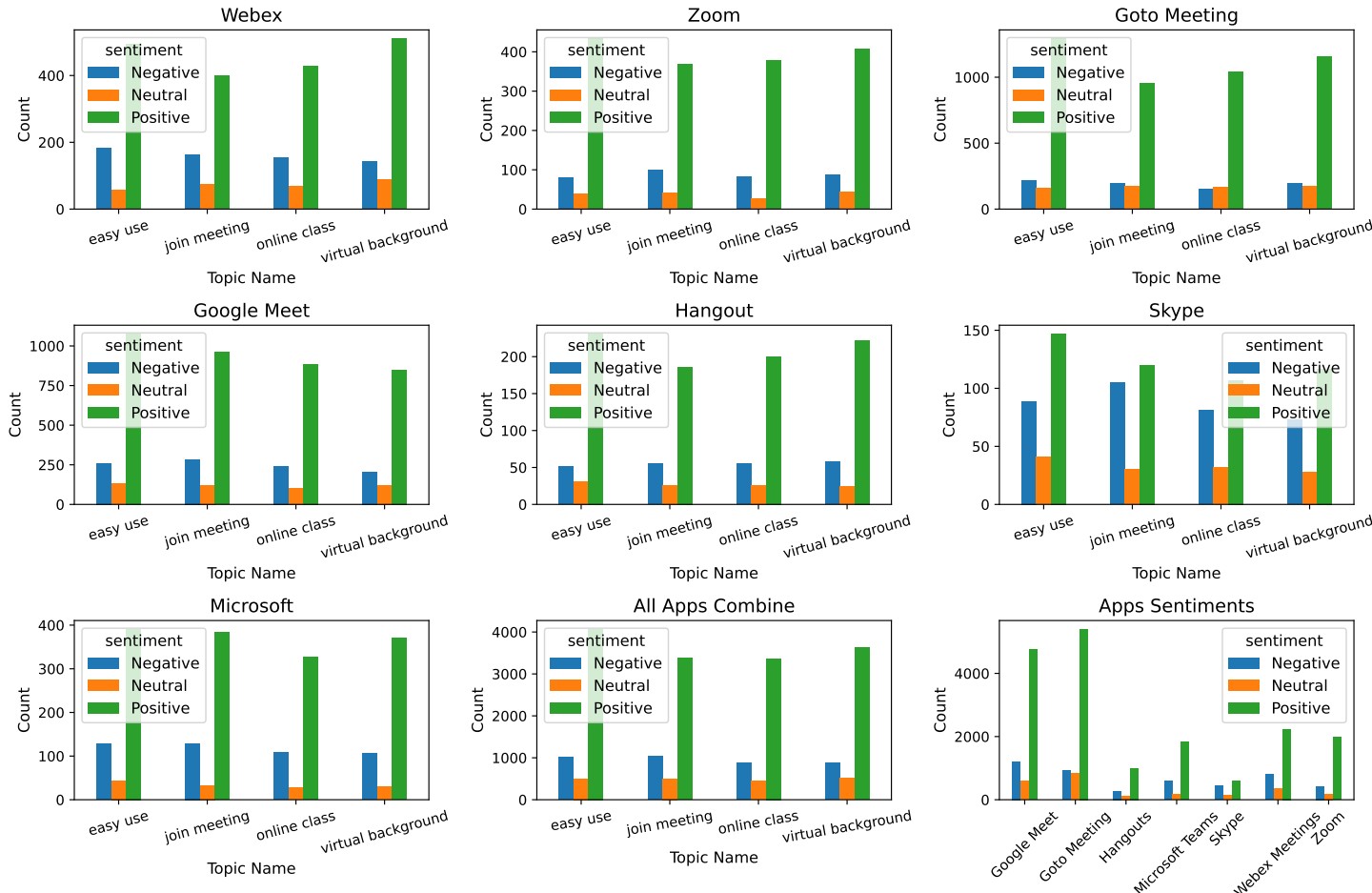

**Figure 6** Topic-wise sentiments count for each app.

topic counts and sentiments for each topic in Fig. 6. It shows that the majority of the positive comments are posted for ease of use for the online meeting apps followed by the virtual background provided by these apps. Although the ratio of negative sentiments is approximately three times low as compared to positive sentiments, most of the negative sentiments are given to joining meetings and ease of use attributes.

The patterns of sentiments for different topic is almost similar for all the apps under discussion; the distribution of topics discussed may slightly vary. Similarly, the positive and negative words used for different apps may vary as well. For example, the negative words used for the Google Meet app are horrible, sad, weak, irritated, etc.

Sentiments for common topics discussed for the Zoom app indicate that the ratio of negative sentiments for topics is slightly less than in the Google Meet app. Similarly, the number of positive words is less comparatively and negative words are slightly different such as sorry, awful, terrible, etc.

Topic sentiments and negative and positive words used for the Goto meeting app indicate that the number of topic sentiments is substantially higher than in Zoom and

Google Meet apps. The ratio of negative topic sentiments is also low than both Zoom and Google Meet apps. The pattern of negative word usage is almost similar to other apps.

Skype-related topic sentiments are very low as compared to other apps and the ratio of negative sentiments is substantially high. The patterns for positive and negative words are similar to other apps. For the Webex app, the number of sentiments is low as compared to other Zoom, and Google Meet apps, it shows a higher ratio of positive sentiments.

In the end, the topics-related sentiments and top words for the Microsoft team and Hangout apps are given. They have a low number of sentiments and a low ratio of negative sentiments for the discussed topics. Similarly, the used negative words are also slightly different than other apps like nasty, regret, and uncomfortable for Hangouts and atrocious, scary, and confusion for the Microsoft team app.

Existing studies report the superior performance of ensemble models over stand-alone machine learning models. So, this study adopts an ensemble approach for sentiment analysis of online meeting apps which have been prevalent recently, especially during the COVID-19 breakout. Traditional ensemble models merge heterogeneous models to get the best of them for obtaining higher performance. Contrary to this approach, this study makes an ensemble model out of a single model. Empirical findings show that the same model shows different performances concerning a feature vector used for training. So this study follows a feature-centric approach and different best-performing features are selected to train the same model. For this purpose, the SVM model is trained using TF-IDF, BoW, and hashing features for sentiment analysis. Experiments are performed using a large dataset of reviews for online meeting apps.

Results demonstrate that the self-voting model tends to improve the performance of stand-alone models. The performance of the models is enhanced regarding two important aspects. First, traditional ensembles use multiple models with a single feature vector for the most part. Although, the advantage of multiple models is obtained, the potential of multiple features is lost. Also, different models may not be suitable for the same data, and combing them may not be prudent. Secondly, it is more rational to use a single model with multiple features if it is performing well on data. Following this rationale, we utilized variants of a single model which are trained using different feature vectors and obtain superior performance. The performance of the self-voting models is much better than single models.

## CONCLUSION

Online meeting apps have been widely used during the COVID-19 pandemic where physical meetings and office work were restricted due to social distancing constraints. A large number of online meeting apps compete by offering a set of unique functions. These apps strive for higher user satisfaction and continue to improve their services in the light of user feedback. The feedback is often posted on the Google app store as views and comments and can be used to perform sentiment analysis for analyzing users' feedback. For accurate sentiment analysis, this study presents a novel concept of self-voting where multiple variants of the same model are trained; each fed with different features. For validation, SVM, DT, LR, and KNN are used with BoW, TF-IDF, and hashing features

on the dataset. Experimental results suggest that the self-voting classification approach elevates the performance of traditional machine learning models. It obtains the accuracy score of 1.00 and 0.98 using hard voting and soft voting, respectively, with the proposed self-voting approach. Reviews analysis indicates that the distribution of positive and negative sentiments for each app varies significantly. For most of the apps, the ratio of positive sentiments is higher than negative sentiments, except for Skype where the ratio is almost similar.

Analysis indicates that predominantly the positive comments appreciate the apps regarding ease of use, and the virtual background provided by these apps. The ratio of negative sentiments is approximately three times low as compared to positive sentiments, and most of the negative sentiments are given to problems in joining meetings and complications in the use of different attributes. This information can be very helpful for online meeting apps to fix these problems to obtain high user quality of service. This study performed analysis for meeting apps; feature-wise analysis was not conducted and should be performed in the future. We also plan to consider deep learning models in the SVM approach and will also consider the imbalanced dataset problem in our future work.

### Funding
This work was supported by the National Natural Science Foundation of China under Grant U1813222, the Tianjin Natural Science Foundation under Grant 18JCYBJC16500, and by the Key Research and Development Project from Hebei Province under Grant 19210404D. There was no additional external funding received for this study. The funders had no role in study design, data collection and analysis, decision to publish, or preparation of the manuscript.

### Grant Disclosures
The following grant information was disclosed by the authors:
The National Natural Science Foundation of China: U1813222.
The Tianjin Natural Science Foundation: 18JCYBJC16500.
The Key Research and Development Project from Hebei Province: 19210404D.

### Competing Interests
Imran Ashraf is an Academic Editor for PeerJ.

### Author Contributions
- Naila Aslam conceived and designed the experiments, analyzed the data, prepared figures and/or tables, and approved the final draft.
- Kewen Xia conceived and designed the experiments, analyzed the data, prepared figures and/or tables, and approved the final draft.
- Furqan Rustam conceived and designed the experiments, performed the computation work, authored or reviewed drafts of the article, and approved the final draft.
- Ernesto Lee performed the experiments, performed the computation work, authored or reviewed drafts of the article, and approved the final draft.
- Imran Ashraf performed the experiments, performed the computation work, authored or reviewed drafts of the article, and approved the final draft.

## Data Availability

The implementation code and raw data are available in the Supplemental Files.

## Supplemental Information

Supplemental information for this article can be found online at http://dx.doi.org/10.7717/peerj-cs.1141#supplemental-information.

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
