# Peer review of "Self voting classification model for online meeting app review sentiment analysis and topic modeling"

_PeerJ Computer Science, doi:10.7717/peerj-cs.1141_

## Round 0.1 · original submission · Major Revisions

We have now received comments from the reviewers. Both the reviewers agree that the work can bring value. However, there are certain aspects that can be improved. Hence, we are making a decision for revision of the manuscript. The decision is based on the reviews received from the reviewers.

·

Basic reporting

I thank the authors for a meticulous clear layout of the introduction and discussion. The background sets the field for the research in question. It highlights the need for the need of a new method of sentiment analysis. I believe this is the focus of the manuscript rather than the mere results of the review of the online apps.

Experimental design

I believe the research question could be improved to highlight the values of this research as a new way of analysis. A secondary outcome is the actual results of the online app reviews. I believe this manuscript is more powerful for other researchers who work in data analysis, and machine learning rather than researchers who work in the social behavioral field.

Validity of the findings

The authors have provided an adequate analysis of the comparison of the proposed new method as compared to other models.

The conclusion is very concise and the authors could elaborate more.

Reviewer 2 ·

Basic reporting

All of the comments and suggestions are available in the additional comments section.

Experimental design

no comment

Validity of the findings

no comment

Additional comments

The idea of self-voting classification is proposed in this paper, in which multiple variants of the same model are trained using alternative feature extraction methodologies, and the final prediction is based on the ensemble of these variants. The authors attempted to deliver a good study, however I have the following issues about accepting it in its current form for publication.

> The abstract and introduction are well written, and the contributions are briefly summarised in the introduction as well. Describe briefly the research's significance.
> The text is repeated and reused in several areas of the paper, and I suggest that the duplicate text be removed.
> In the methodology, describe the model explicitly. Detailed description is required about the data.
> I would advise against describing the procedures employed. The authors should explain how and why they employed the existing methodologies. I noticed this in both the pre-processing and prior portions.
> Already known equations are utilised throughout the paper to demonstrate how the suggested methodology works, although I oppose presenting them in this paper. Authors should concentrate on their model.
> Try to decrease the number of figures in the result section; if the behaviour of the figures is nearly same, authors can merge them.
> At the end of the result section, provide a genuine discussion.
> The true conclusion of the paper should be provided by the authors. The majority of the conclusion is currently a summary of the paper.
> Authors can present their framework using a flowchart or algorithm, with less emphasis on existing methodologies.

---

## Round 0.2 · Minor Revisions

Both the reviewers have identified that minor revisions are required. The authors are advised to carefully address the comments. For example, both the reviewers have identified that some of the figures are redundant and can be removed. Both the reviewers have mentioned that there is a need for English proofreading.

·

Basic reporting

Overall, I am satisfied with the authors' response.

There is still room for some English editing of some of the sentences.

Experimental design

I have no further comments

Validity of the findings

There is still room for modification of the discussion and conclusion. The conclusion is still part of the discussion and not setting the ground for further research or implications of the current study findings.

Additional comments

I agree with reviewer 2 that the figures about each app are not needed in the manuscript especially that the focus is on the model and not just the results of the sentiment analysis. They are mostly repetitive and show similar results. The results can be summarize in the text and refer to an Appendix that contains al the figures.

Reviewer 2 ·

Basic reporting

no comment

Experimental design

More explanation is needed for Figures 6–12. The size of the paper can be reduced by grouping the Figures.

Validity of the findings

no comment

Additional comments

- Take off the title "Discussion," the body paragraphs have sufficient detail.
- Add details (step by step) to the algorithms

---

## Author Rebuttal · Round 0.2

# Response to Comments

**Manuscript ID**: CS-2022:07:75341:1:0:NEW

**Title**: Self voting classification model for online meeting app review sentiment analysis and topic modeling

**Authors**: Naila Aslam, Kewen Xia, Furqan Rustam, Ernesto Lee and  Imran Ashraf

Dear Editor,

Thank you very much for allowing us to revise the manuscript. We would like to thank the editor and all the reviewers for their valuable comments and suggestions. Based on the feedback, we have revised our manuscript. The detailed modifications to address reviewers' comments are provided in the following. For clarity, we have marked our responses in blue. Whenever we copy a paragraph from the manuscript here, we mark it as a red color.

## Answers to Comments

**REVIEWER 1**

**Comment 1:** I thank the authors for a meticulous clear layout of the introduction and discussion. The background sets the field for the research in question. It highlights the need for the need of a new method of sentiment analysis. I believe this is the focus of the manuscript rather than the mere results of the review of the online apps.

**Response:** We would like to thank the reviewer for appreciating the study. We focused on machine learning and app reviews analysis.

**Comment 2:** I believe the research question could be improved to highlight the values of this research as a new way of analysis. A secondary outcome is the actual results of the online app reviews. I believe this manuscript is more powerful for other researchers who work in data analysis, and machine learning rather than researchers who work in the social behavioral field.

**Response:** Thank you very much for your suggestions. We refined the contributions of this research as suggested by the worthy reviewer.

- The study performs sentiment analysis of tweets for online meeting apps using a novel self-voting ensemble model. The self-voting model combines three variants of the same model, however, the features fed to each model are different. Performance is analyzed using both the hard voting and soft voting criteria.
- For performance analysis, support vector machine (SVM), decision tree (DT), logistic regression (LR), and k nearest neighbor (KNN) models are used with three different feature extraction approaches including term frequency-inverse document frequency (TF-IDF), the bag of words (BoW), and hashing.
- For experiments, a large dataset of online meeting apps tweets has been collected. Dataset is labeled using the valence aware dictionary for sentiment reasoning (VADER) while for topic modeling, the latent Dirichlet allocation (LDA) approach is used. Performance is evaluated using accuracy, precision, recall, and F1 score. In addition, a comparison of the proposed model is carried out with the state-of-the-art approaches.

**Comment 3:** The authors have provided an adequate analysis of the comparison of the proposed new method as compared to other models.

**Response:** We would like to thank the reviewer for the appreciation.

**Comment 4**: The conclusion is very concise and the authors could elaborate more.

**Response:** We updated the conclusion section according to reviewer's suggestion.

Online meeting apps have been widely used during the COVID-19 pandemic era where physical meetings and office work were restricted due to social distancing constraints. A large number of online meeting apps compete by offering a set of unique functions for higher user satisfaction and continue to improve their services in the light of user feedback. The feedback is often posted on the Google app store as views and comments and requires efficient analysis, where sentiment analysis comes in handy. For accurate sentiment analysis, this study presents a novel concept of self-voting where multiple variants of the same model are trained; each fed with different features. For validation, SMV, DT, LR, and KNN are used with BoW, TF-IDF, and hashing features on the dataset containing user reviews of online meeting apps. Experimental results suggest that the self-voting classification approach elevates the performance of traditional machine learning models. Contrary to stand-alone models, a self-voting ensemble is more influential to obtain higher accuracy. For the task at hand, SVM obtains the accuracy score of 1.00 and 0.98 using hard voting and soft voting, respectively, with the proposed self-voting approach. Results show that different features show different accuracy for individual classes like positive, negative, and neutral. Combing the features for a

single model is a better choice which substantially improves the overall performance of a model. Performance comparison with existing studies shows that the proposed approach outperforms these models. In future work, we intend to consider deep learning models in the SVM approach and will also consider the imbalanced dataset problem in our future work.

**REVIEWER 2**

**Comment 1:** The abstract and introduction are well written, and the contributions are briefly summarised in the introduction as well. Describe briefly the research's significance

**Response:** We would like to thank the reviewer for valuable comments. In this study, we perform analysis for online meeting apps and proposed an approach for sentiment analysis. This analysis can help online meeting apps owner to improve their app quality to attract more users. We modified the contributions of the study, as per the suggestions of the worthy reviewer.

- The study performs sentiment analysis of tweets for online meeting apps using a novel self-voting ensemble model. The self-voting model combines three variants of the same model, however, the features fed to each model are different. Performance is analyzed using both the hard voting and soft voting criteria.
- For performance analysis, support vector machine (SVM), decision tree (DT), logistic regression (LR), and k nearest neighbor (KNN) models are used with three different feature extraction approaches including term frequency-inverse document frequency (TF-IDF), the bag of words (BoW), and hashing.
- For experiments, a large dataset of online meeting apps tweets has been collected. Dataset is labeled using the valence aware dictionary for sentiment reasoning (VADER) while for topic modeling, the latent Dirichlet allocation (LDA) approach is used. Performance is evaluated using accuracy, precision, recall, and F1 score. In addition, a comparison of the proposed model is carried out with the state-of-the-art approaches.

**Comment 2:** The text is repeated and reused in several areas of the paper, and I suggest that the duplicate text be removed.

**Response:** The authros are highly grateful for your valuable insights. We updated the manuscript to remove duplications.

**Comment 3:** In the methodology, describe the model explicitly. Detailed description is required about the data.

**Response:** We added more descriptions about the used dataset in the dataset description sections while the description of the model is present in the section (Self-Voting Classifier).

**Dataset Description**

The dataset is extracted from the Google play store for several online meeting apps including 'Google Meet', 'Goto Meeting', 'Zoom Meeting', 'Skype', 'Hangouts', 'Microsoft Teams', and 'Webex Meeting'. These apps have been selected regarding their overall rating on the Google play store. The app's reviews are extracted for the period of 12 October 2018 to 7 December 2021. This study considers only the reviews given in the English language. The reviews are collected using the Google play scraper library. The collected dataset contains the review id, user name, the content of reviews, score by user for the app, thumbs up count, review created version, and data for the review posted. The reviews contain opinions of users regarding particular positive and negative features of an app. Besides criticism, such reviews also contain suggestions for improvement or the addition of new features. These reviews are very helpful for app companies to make changes according to user sentiments. Sample data from the collected dataset is shown in Table 1. The number of reviews varies for each app and the distribution of reviews is provided in Figure 2.

**Comment 4:** I would advise against describing the procedures employed. The authors should explain how and why they employed the existing methodologies. I noticed this in both the pre-processing and prior portions.

**Response:** Thank you very much for your suggestions. The description of employed machine learning models is removed from the revised manuscript. The description of the proposed approach is enhanced.

**Self-Voting Classifier**

This study proposes a novel voting classifier, called a self-voting classifier. Traditional ensemble models follow a group voting mechanism, using heterogeneous models where the output of multiple models is combined using soft or hard voting criteria. Since the performance of different models vary, combining the prediction of multiple models improves the classification performance (23,19,18). Contrary to the group voting from heterogeneous models, this study adopts the self-voting ensemble where the output of the three different variants of SVM is combined to make the final prediction. Since the performance of a model varies with respect to the features fed for training, the idea is to feed multiple features to the same model and combine them to make the ensemble. Three SVM variants have been trained on different feature vectors including BoW, TF-IDF, and hashing features. The performance of self voting approach is investigated both using the soft and hard voting criteria.

**Comment 5:** Already known equations are utilised throughout the paper to demonstrate how the suggested methodology works, although I oppose presenting them in this paper. Authors should concentrate on their model.

**Response:** Respected reviewer, we have utilized the existing equations since we are using exising SVM model and other machine learning models. Equations are used to present the working of the proposed approach. Since SVM is alreaded well presented in the literature, we do not present its working mechanism. We present the details of how multiple features are combined for SVM to make the final decision using self voting approach.

**Comment 6:** Try to decrease the number of figures in the result section; if the behaviour of the figures is nearly same, authors can merge them.

**Response:** Reviewer concern is valuable but the figures are for different apps. In addition, due to short space, combining figures under subfigure would make it very difficult for the readers to understand.

**Comment 7:** At the end of the result section, provide a genuine discussion.

**Response:** Discussion section has been added to the revised manuscript.

## Discussions

Existing studies report the superior performance of ensemble models over stand-alone machine learning models. So, this study adopts an ensemble approach for sentiment analysis of online meeting apps which have been prevalent recently, especially during the COVID-19 breakout. Traditional ensemble models merge heterogeneous models to get the best of them for obtaining higher performance. Contrary to this approach, this study makes an ensemble model out of a single model. Empirical findings show that the same model shows different performance with respect to a feature vector used for training. So this study follows a feature-centric approach and different best-performing features are selected to train the same model. For this purpose, the SVM model is trained using TF-IDF, BoW, and hashing features for sentiment analysis. Experiments are performed using a large dataset of reviews for online meeting apps.

Results demonstrate that the self-voting model tends to improve the performance of stand-alone models. The performance of the models is enhanced regarding two important aspects. First, traditional ensembles use multiple models with a single feature vector for the most part. Although, the advantage of multiple models is obtained but the potential of multiple features is lost. Also, different models may not be suitable for the same data, and combing them may not be prudent. Secondly, it is more rational to use a single model with multiple features if it is performing well on data. Following this rationale, we utilized variants of a single model which are trained using different feature vectors and obtain superior performance. The performance of the self-voting models is much better than single models.

**Comment 8:** The true conclusion of the paper should be provided by the authors. The majority of the conclusion is currently a summary of the paper.

**Response:** The conclusion section is modified.

Online meeting apps have been widely used during the COVID-19 pandemic era where physical meetings and office work were restricted due to social distancing constraints. A large number of online meeting apps compete by offering a set of unique functions for higher user satisfaction and continue to improve their services in the light of user feedback. The feedback is often posted on the Google app store as views and comments and requires efficient analysis, where sentiment analysis comes in handy. For accurate sentiment analysis, this study presents a novel concept of self-voting where multiple variants of the same model are trained; each fed with different features. For validation, SMV, DT, LR, and KNN are used with BoW, TF-IDF, and hashing features on the dataset containing user reviews of online meeting apps. Experimental results suggest that the self-voting classification approach elevates the performance of traditional machine learning models. Contrary to stand-alone models, a self-voting ensemble is more influential to obtain higher accuracy. For the task at hand, SVM obtains the accuracy score of 1.00 and 0.98 using hard voting and soft voting, respectively, with the proposed self-voting approach. Results show that different features show different accuracy for individual classes like positive, negative, and neutral. Combing the features for a single model is a better choice which substantially improves the overall performance of a model. Performance comparison with existing studies shows that the proposed approach outperforms these models. In future work, we intend to consider deep learning models in the SVM approach and will also consider the imbalanced dataset problem in our future work.

**Comment 9:** Authors can present their framework using a flowchart or algorithm, with less emphasis on existing methodologies.

**Response:** The algorithm shows the step to propose SVM-SV model. Here SVM_T is trained using TF-IDF features while SVM_B and SVM_H are trained using BoW and Hashing features. P1, P2, and P3 are the predictions by trained SVM models on the test data.

**Algorithm 1** Proposed SVM-SV algorithm

   **Input:** Apps Reviews
   **Output:** $Positive|Negative|Neutral$

1:   Def Model_Training():
2:     $SVM_T \leftarrow$ SVM(TF-IDF_Features)
3:     $SVM_B \leftarrow$ SVM(BoW_Features)
4:     $SVM_H \leftarrow$ SVM(Hashing_Features)
5:   **for** $i$ in Test_Corpus **do**
6:     $P1 \leftarrow SVM_T(i)$
7:     $P2 \leftarrow SVM_B(i)$
8:     $P3 \leftarrow SVM_H(i)$
9:     $SVM-SV(Pred) \leftarrow mode\{P1,P2,P3\}$
10:   **end for**
11:   $Positive|Negative|Neutral \leftarrow SVM-SV$ prediction
* * *
**Algorithm 2** Proposed SVM-HV algorithm

   **Input:** Apps Reviews
   **Output:** $Positive|Negative|Neutral$

1:   Def Model_Training():
2:     $SVM_T \leftarrow$ SVM(TF-IDF_Features)
3:     $SVM_B \leftarrow$ SVM(BoW_Features)
4:     $SVM_H \leftarrow$ SVM(Hashing_Features)
5:   **for** $i$ in Test_Corpus **do**
6:     $Pos1 \leftarrow SVM_T(i)$
7:     $Neg1 \leftarrow SVM_T(i)$
8:     $Neu1 \leftarrow SVM_T(i)$
9:     $Pos2 \leftarrow SVM_B(i)$
10:     $Neg2 \leftarrow SVM_B(i)$
11:     $Neu2 \leftarrow SVM_B(i)$
12:     $Pos3 \leftarrow SVM_H(i)$
13:     $Neg3 \leftarrow SVM_H(i)$
14:     $Neu3 \leftarrow SVM_H(i)$
15:     $Prob\_Pos \leftarrow \frac{(Pos1+Pos2+Pos3)}{3}$
16:     $Prob\_Neg \leftarrow \frac{(Neg1+Neg2+Neg3)}{3}$
17:     $Prob\_Neu \leftarrow \frac{(Neu1+Neu2+Neu3)}{3}$
18:     $SVM-HV(Pred) \leftarrow argmax\{Prob\_Pos,Prob\_Neg,Prob\_Neu\}$
19:   **end for**
20:   $Positive|Negative|Neutral \leftarrow SVM-HV$ prediction

---

## Round 0.3 · accepted · Accept

I am pleased to accept the paper. The topic is interesting for the readers and we can hope for a good discussion around the topic.